# Legally Enforceable Hate Speech Detection for Public Forums

**Chu Fei Luo** [1,2], **Rohan Bhambhoria** [1,2], **Samuel Dahan**[2,3], **and Xiaodan Zhu**[1,2]

[1]Department of Electrical and Computer Engineering & Ingenuity Labs Research Institute
Queen's University
[2]Conflict Analytics Lab, Queen's University
[3]Cornell Law School

{chufei.luo,r.bhambhoria,xiaodan.zhu,samuel.dahan}@queensu.ca

## Abstract

Hate speech causes widespread and deep-seated societal issues. Proper enforcement of hate speech laws is key for protecting groups of people against harmful and discriminatory language. However, determining what constitutes hate speech is a complex task that is highly open to subjective interpretations. Existing works do not align their systems with enforceable definitions of hate speech, which can make their outputs inconsistent with the goals of regulators. This research introduces a new perspective and task for enforceable hate speech detection centred around legal definitions, and a dataset annotated on violations of eleven possible definitions by legal experts. Given the challenge of identifying clear, legally enforceable instances of hate speech, we augment the dataset with expert-generated samples and an automatically mined challenge set. We experiment with grounding the model decision in these definitions using zero-shot and few-shot prompting. We then report results on several large language models (LLMs). With this task definition, automatic hate speech detection can be more closely aligned to enforceable laws, and hence assist in more rigorous enforcement of legal protections against harmful speech in public forums. [1]

## 1 Introduction

Hate speech is regarded as "a denial of the values of tolerance, inclusion, diversity and the very essence of human rights norms and principles."[2] The internet and public forums are global assets that facilitate interaction between people, but the ease of communication also enables hate speech to travel rapidly and spread on a large scale, causing serious societal and security issues (Rapp, 2021; Alkiviadou, 2019). To ensure compliance with hate

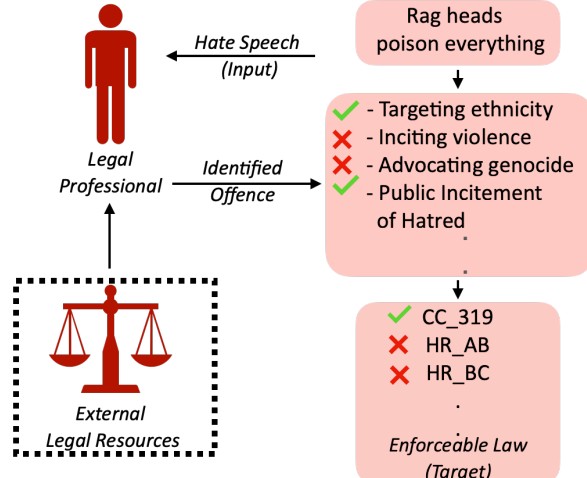

Figure 1: A visualization of our proposed method to ground hate speech to specialized legal definitions. A legal professional reads external legal resources and makes a judgement on some hate speech input, then identifies offences according to our definitions and makes a judgement on violations.

speech laws, we argue that automatic hate speech detection is essential for monitoring these forums due to their large scale. This work analyzes the effectiveness of machine learning methods on *enforceable* hate speech laws, where enforceable is defined as a law or rule that is "possible to make people obey, or possible to make happen or be accepted."[3] We believe the advancement of artificial intelligence can be utilized to tackle hate speech by better aligning with regulations that protect groups of people against harmful discrimination, and hence help build a healthy society.

Definitions of hate speech can vary to the point that two datasets are inconsistent with each other (Khurana et al., 2022; Fortuna et al., 2020). Possibly due to these divergences in hate speech definitions, previous works find that models trained on one dataset often perform poorly when evalu-

---

[1]The code for this paper is publicly available at https://github.com/chufeiluo/legalhatespeech

[2]https://www.un.org/en/hate-speech/impact-and-prevention/why-tackle-hate-speech

[3]https://dictionary.cambridge.org/dictionary/english/enforceable

ated on another (Yin and Zubiaga, 2021; Kim et al., 2022). Sachdeva et al. (2022) note this is due to intersubjectivity in annotated datasets — there is high variability in annotator opinions on highly complex tasks, including hate speech detection. This subjectivity is not specific to computer science; hate speech is a highly debated topic, with definitions varying significantly across regions and organizations (Brown, 2015; Zufall et al., 2022). Previous works try to remove these variations with modelling strategies like contrastive learning (Kim et al., 2022) or data perspectivism for de-biasing (Sachdeva et al., 2022). However, they define hate speech concepts that have not been explicitly aligned to legal definitions. When models are not trained with awareness of the law, the output is likely irrelevant and not in line with the accepted and enforceable legal definitions under concern.

In this work, we introduce a new task formulation grounded in eleven definitions of hate speech curated from criminal law, human rights codes, and hateful conduct policies from public forums. We provide a gold label dataset for violations of these hate speech definitions, annotated by legal experts. Statements that could lead to punitive consequences are generally rare, so we augment the positive class with *edited samples* from our expert annotators, and employ data filtering methods to increase the size. Due to the expense of domain expert annotation, instruction-tuned LLMs are a reasonable candidate for hate speech detection. They have shown promising results on legal tasks with prompting (Guha et al., 2022), and increasingly long context windows allow models to process longer pieces of text (OpenAI, 2023). We report baseline performance using our definitions to prompt the state-of-the-art LLMs, and we also use parameter-efficient tuning (Taori et al., 2023; Hu et al., 2021) and self training to improve the performance of smaller LLMs. Additionally, we construct a silver dataset with automatically generated labels. The contributions of our research are summarized as follows:

- We propose a new perspective and task for hate speech detection grounded in legal definitions to adhere with accepted enforceable standards.

- We built a novel dataset for this task, with annotations for eleven legal definitions across several jurisdictions and platforms. We provide a gold dataset annotated by legal experts, a silver

dataset with automatically generated labels, and a challenge dataset for unsupervised methods such as self training.

- Empirical baselines on the state-of-the-art large language models (LLMs) have been established to facilitate further studies on the problem. We report results with task instruction prompting, zero-shot prompting, and parameter-efficient finetuning (PEFT).

## 2 Related Work

**Hate speech** Hate speech has significant societal impact. Automatic hate speech detection is essential in regulating large online communities such as social media (Yin and Zubiaga, 2021). However, there are significant issues with dataset quality (Khurana et al., 2022; Fortuna et al., 2020; Yin and Zubiaga, 2021) due to the inherent subjectivity of hate speech (Sachdeva et al., 2022; Brown, 2015). Hate speech is also highly complex, and various works choose to focus on one component. ElSherief et al. (2021) target implicit hate speech, and Yu et al. (2022) demonstrate that context can often change the degree of hate in a statement.

There are several strategies to mitigate intersubjectivity, forming three approximate groups: task definition, model training, and data pre-processing. For task definition, the research introduced in Khurana et al. (2022); Zufall et al. (2022) is the only work to our knowledge that defines a detailed annotation framework in collaboration with legal experts and social scientists. Zufall et al. (2022) propose a framework around the EU's definition of hate speech, which is the minimum standard in Europe. Modelling strategies like contrastive learning (Kim et al., 2022) reduce variance in the embeddings, and few-shot methods reduce requirements for training data (AlKhamissi et al., 2022). Finally, Sachdeva et al. (2022) propose a data-level de-biasing method to reduce biases between annotators based on their demographics. In contrast, our work attempts to reconcile multiple legal definitions of hate speech within one jurisdiction.

**Prompting** Prompting has shown great success in zero-shot and few-shot settings (Brown et al., 2020). Although there are concerns to the efficacy of prompts (Webson and Pavlick, 2022; Khashabi et al., 2022), incorporating natural language instructions in the input text is a common strategy to improve zero-shot performance (Chen et al., 2021).

Guha et al. (2022) show LLMs can obtain reasonable performance based on task definitions for specialized tasks in law.

With the emergence of instruction-tuned language models trained on human feedback, large language models have seen emergent capabilities in zero-shot settings with instruction prompting (Ouyang et al., 2022; OpenAI, 2023). However, there is a distinction between a model's ability to understand language and reason (Mahowald et al., 2023). Early studies show that large language models are still insufficient on many high-stakes tasks (Li et al., 2023). Additionally, the imprecision of natural language outputs often hinders non-generative tasks like classification. Hallucinations occur when a model is uncertain of its decision but does not know how to express itself, or does not have proper reasoning for the task (Mukherjee et al., 2023). To this end, many works attempt closer supervision of the model's thought process, including chain-of-thought prompting (Wei et al., 2022) and process supervision (Lightman et al., 2023).

**Parameter-Efficient Tuning** Many previous studies observe that a fine-tuned model performs better than zero-shot or few-shot prompting (Li et al., 2023). LLMs like ChatGPT (Ouyang et al., 2022) or LLAMA (Touvron et al., 2023) require significant computational resources to update parameters. This makes parameter-efficient tuning methods highly desirable to increase the efficiency of model training. LLMs such as LLAMA with 7 billion parameters (Taori et al., 2023), can be tuned on consumer hardware with Low-Rank Adaptation (LoRA) (Hu et al., 2021).

## 3 Enforceable Hate Speech Detection

Our research aims to detect hate speech statements that violate *at least one* legally enforceable definition of hate speech, which we call **Enforceable Hate Speech**. This is illustrated in Figure 1. We utilize eleven definitions of hate speech from three legal sources, which are applicable to language and conduct published on public forums. For legal details, please refer to Appendix A.1.

- **Criminal Code**: Federal laws that are established by a country's legislature and codifies the country's criminal offences. Violations can result in criminal charges. In this work, we use the Canadian Criminal Code, but the framework can be easily extended.

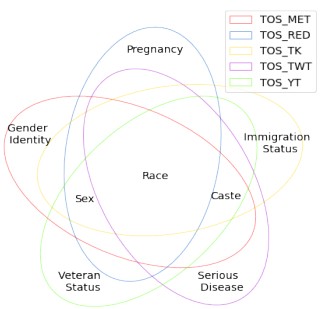

Figure 2: Taking our five social media policies as examples, we illustrate the overlaps and differences between various protected groups. Most definitions have at least one unique protected group.

- **Human Rights Code**: Some jurisdictions have human rights codes separate from criminal laws. While these infractions do not typically lead to incarceration, violations often result in monetary compensation for victims. We collected codes from four Canadian provinces and territories. Again, the research approach can be easily extended to other code.

- **Hateful Conduct Policies**: Terms and Conditions for content posted to social media. Violations can result in removal of the problematic content, and suspension/removal of the account in serious cases. We collected policies on hateful conduct from five social media platforms, as of December 2022.

Each definition is comprised of two components: a *description* of unacceptable behaviour or conduct, and a list of *protected groups* under their policy. Human Rights Codes describe a violation as any statement "likely to expose an individual or class of individuals to hatred or contempt," whereas Hateful Conduct Policies protect against content that "targets a protected group." While these are phrased distinctly, the offending behaviours detailed in these policies are very similar to each other, so the primary distinction between most of our definitions is in their **protected groups**. As illustrated in Figure 2, there are more common protected groups, like Race, Sex/Gender, or Caste (family status), and rarer groups like Veteran Status or Serious Disease. The Criminal Code describes more severe offences of promoting genocide or public incitements of hatred that are "likely to lead to a breach in peace."

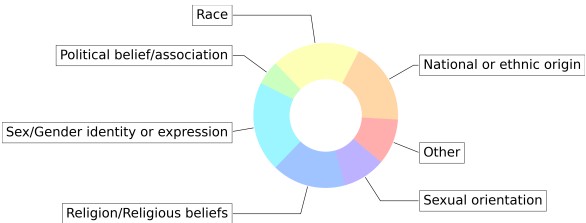

Figure 3: Diagram showing frequency of protected groups that appear in our dataset, as annotated by legal experts.

## 3.1 Data collection

Our dataset contains samples from existing datasets with the aim to determine how previous annotations align with legal definitions of hate speech. Additionally, doing so gains greater semantic variety as we sample from a diverse set of public forums. Next, we obtain annotations from legal experts. We employ editing annotation tasks and data filtering to increase the size of our gold data, while utilizing experts to obtain high-quality labels with detailed reasoning. Finally, we further sample a silver dataset with automatically generated labels.

**Non-enforceable Datasets.** We use five English hate speech and toxicity datasets released in the public domain.

- **CivilComments** (Borkan et al., 2019) was sourced from the Civil Comments database, a comment widget that allowed peer-reviewed comments on news sites. We randomly mined comments from the CivilComments train dataset that had some toxicity value above 0.5.
- **SBIC** (Sap et al., 2020) was obtained from Gab and Twitter. They do not have direct labels for hate speech, but we infer a heuristic for positive cases based on previous works (AlKhamissi et al., 2022) — that is, offensive language that targets a group is considered a positive sample.
- **HateXplain** (Mathew et al., 2021) was created from Twitter for the goal of evaluating interpretability methods.
- **Implicit Hate Speech** (**IHS**) (ElSherief et al., 2021) was collected from Twitter, including fine-grained labels for various types of hate speech.
- **CounterContext** (Yu et al., 2022) was acquired from Reddit, by collecting pairs of samples to provide more context within the public forum.

The distribution of our dataset is shown in Table 1. For further details on public forum statistics,

please refer to Appendix A.3. There were four samples of identical text from different sources, likely cross-posted between Reddit and Twitter, but they only appear once in our dataset, and we confirm their original datasets give these four overlapping samples the same annotation labels. For this reason, we count one positive sample twice in Table 1.

**Enforceable Annotations.** We collaborated with ten volunteer legal experts over two rounds of annotations. They are law school graduate students with an average of two years of education into their J.D. degree. We distributed posts to each expert in batches, such that each post would have at least three unique experts, and obtain a label with majority voting. While there are many clear statements of hateful speech, there are also times when the speaker's original meaning cannot be discerned, but the phrase contains hateful language or ambiguous implications that an expert believes would require further context. Hence, we provide an Unclear option for three possible classes: Violates, Does not Violate, and Unclear. Additionally, we ask the annotator to identify protected groups being targeted, and provide their reasoning for a violation in free-form text. We employ an or-wise combination of all annotators, where a group is selected if at least one annotator believes they were targeted. Then, the annotators mark the specific definitions violated. For detailed annotator instructions, refer to Appendix A.2.

After the first round of annotations, there were 488 total hate speech statements, with 22 (4.5%) samples in the positive class, and 28 (5.7%) labelled Unclear. We note that our positive rate is lower than prior datasets over all source datasets except for HateXplain. Some previous datasets such as IHS and CounterContext focus micro-aggressions and implicit statements, which are the main focus of Implicit Hate Speech and Counter-Context. We argue these works are not aligned to the legal concept of hate speech, instead capturing the broader concept of discrimination (prejudice towards certain groups of people). We internally discussed the enforceability of discrimination versus hate speech at great lengths before removing discrimination policies from our definition. Discrimination is a broad concept and its impact on society is varied, making the legal implications not well defined.

The following is a label-flipped sample from Implicit Hate Speech:

"this is crazy. how do you give a loan to someone who can be deported at will ?"

The speaker believes "someone who can be deported at will" (immigrants) should not be able to get loans, which is discrimination but not hate speech. Hate speech in the law is defined as directly insulting/dehumanizing/humiliating someone based on a protected attribute. Multiple definitions reference, for example, comparing a group of people to insects. Since previous works do not reference the law, they can conflate different legal concepts, and our understanding of hate speech in the NLP community becomes misaligned with experts in other disciplines.

**Expert Augmentation.** We wish to further scale the size of our data while minimizing requirements for expert annotation. To strengthen the training signal, we asked the annotators to *edit existing posts* in the dataset that were not hate speech such that it would become a violating post. First, we selected posts that either had a minority vote for the positive class, or at least one `Unclear` annotation. We obtain 80 such samples from our first round of annotations, and instructed the annotators to edit the statements to specific definitions. Where they were unsure of how to edit the post, we asked the experts to create new statements.

This process is repeated with 1,100 samples of Reddit data, which is a subset of the automatically obtained data described below. We divided the posts among the annotators, asking them to edit the posts to become hate speech. The annotators noted some difficulty with the editing task, so for the latter half, we asked them to only make edits that changed *less than half* of the statement. For example, replacing a neutral insult such as "jerk" with a slur could change the statement to hate speech. Overall, we obtained 179 edited statements from the experts.

Additional positive samples of hate speech were obtained from `HateXplain`, which had the greatest correlation to our positives from our initial round of annotations. We also re-incorporate our expert edits, along with the expert's own annotation of violating categories. To ensure quality in our edited statements, we obtain two additional annotations per edited sample and perform majority voting. After this process, we have 165 positive samples, as shown in Table 1. The most common groups targeted are illustrated in Figure 3. These groups are protected by all hate speech definitions, except for "Political belief/association."

| Dataset | Count | Pos-o | Pos-l | Uncl. |
|---|---|---|---|---|
| CivComments | 418 | - | 10 (2.3) | 17 (4.5) |
| SBIC | 20 | 9 (45.0) | 2 (10.0) | 1 (5.0) |
| HateXplain | 20 | 8 (40.0) | 10 (50.0) | 7 (13.0) |
| IHS | 20 | 9 (45.0) | 0 (0.0) | 0 (0.0) |
| CounterContext | 10 | 7 (70.0) | 0 (0.0) | 0 (0.0) |
| Total | 488 | 39 (7.9) | 22 (4.5) | 28 (5.7) |
| with Augment. | 704 | - | 165 (23.4) | 38 (5.4) |

Table 1: Dataset distribution by sampling source, as well as comparison of positive rates by previous hate speech definitions (Pos-o) to enforceable legal definitions (Pos-l). Uncl. refers to Unclear label rate. Each entry is reported in the format Value (Rate %)
.

**Automatic Augmentation.** Recent works have shown that the state-of-the-art LLMs are able to generate high-quality labels (Mukherjee et al., 2023), so we extend our dataset with raw Reddit data and silver labels from zero-shot prompting. We scraped Reddit submissions and comments from February 21, 2022 to September 5, 2022. We chose 10 random but popular communities (also known as Subreddits), as well as 10 that were indexed as high toxicity by Rajadesingan et al. (2020). Many of the Subreddits from their work have been removed, but we were able to find 10 active forums with some positive toxicity score. We collected both submission text and comments for 7.8 million posts total. From this set, we sampled 100,000 posts with maximum cosine similarity to at least one of our gold label positive samples $\tau_{max} = 0.55$. We then generate labels with the current state-of-the-art language model, `GPT-4`, as we found it produced high quality labels. We use the prompt template from our experiments, as shown in Table 2, and multi-target prompting described in Section 4.

During this process, we also sample obvious negative samples in a similar way. If the maximum cosine similarity of one sample to all of our gold positives is less than a threshold $\tau_{min} = 0.3$, this sample is saved as an easy negative. We use these easy negatives to increase the size of our gold data, such that our dataset matches the organically collected positive rate. This automatic augmentation results in a gold dataset with 2,204 samples, and a silver dataset with 2,007 samples.

## 4 Methodology

### 4.1 Classification with Legal Reasoning

**Classification.** We consider a baseline of zero-shot classification with prompted instructions.

| Definition | [target] states [description]. Protected groups include [groups]. |
|---|---|
| Classification | Is this post a violation of [target]? Please answer Yes, No, or Unclear. |
| Reasoning | Is this post a violation of [target]? Please answer Yes/No/Unclear. If Yes, explain why. |

Table 2: Templates used for all experiments. [target] is an arbitrary name chosen to strengthen the relationship between the definition and question. [description] and [groups] are components of the definition. One prompt can have multiple definitions.

Given a set of enforceable definitions $D$, we attempt to predict whether a statement violates one of the policies. We perform **coarse-grained** and **fine-grained** classification.

For all experiments, we prompt the model with input text and the target definitions, using a unified template shown in Table 2. To be precise, any generated answer starting with Yes is a positive prediction, and any with No is an explicit negative. Hallucinations, where conditional generation models generate outputs outside of the set of target responses (Ji et al., 2022), is an important measure for model robustness, so we report *hallucination rate* in our responses. We explicitly ask the model to respond with Yes, No, or Unclear, as shown in Table 2, and define a hallucination as any response that *does not begin* with any of these options.

Specifically, we consider the following typical large language models:

- **RoBERTa-Large** (Liu et al., 2019). As a baseline, we train a pre-trained RoBERTa model with 355M parameters. RoBERTa-Large is an encoder-only LLM, while the others have autoregressive encoder-decoder architectures. To obtain comparable zero-shot results, we use the LM-BFF method (Gao et al., 2021).
- **Alpaca-LoRA** (Taori et al., 2023). We use an open-source version of Alpaca, which is LLAMA trained with instruction tuning (Touvron et al., 2023; Ouyang et al., 2022). This version is reproduced with Low-Rank Adaptation (LoRA) (Hu et al., 2021). We use the 7-billion parameter variant of LLAMA as the base model.
- **Falcon** (Penedo et al., 2023). Falcon is an open-source LLM with multi-query attention

heads, trained on 1.5 trillion tokens from a novel dataset, RefinedWeb. We run their 7b model in our experiments.
- **WizardLM** (Choi et al., 2022). This language model was trained on data, refactored with more complex instructions generated by LLMs. Since our task is highly contingent on understanding instructions, we choose their model as a strong candidate baseline. We utilize their 13-billion parameter model.
- **Llama 2** (Touvron et al., 2023). Currently the state of the art in open source language models, this is a model trained with a novel Grouped-Query Attention mechanism and longer context length compared to its predecessor. We use the llama-2-chat-13b checkpoint.
- **GPT-3.5** (Ouyang et al., 2022). We use the gpt-3.5-turbo-0613 model, with 8K and 16K context windows available. These two models were trained with human feedback from GPT-3 with 175B parameters.
- **GPT-4** (OpenAI, 2023). GPT-4 is an autoregressive LLM trained with further human feedback. The exact number of parameters is unknown, but it is known to be larger than GPT-3.5.

**Legal Reasoning.** Increasing points of thought and reasoning steps have been noted to significantly improve performance in LLMs, shown in both model training with emergent papers like process supervision (Lightman et al., 2023) and zero-shot inference like chain-of-thought prompting (Wei et al., 2022). In this work, we prompt the model to explain its classification. We choose to provide minimal guidance in the prompt as a baseline, and to observe how the language model responds to uncertainty. For this setting, we add an additional hallucination criteria where the model does not output any explanation.

We want to examine the quality of the reasoning by mapping it to the definition components, i.e. the *protected group* being targeted. We obtain a bag-of-words representation of the response as a set of the word tokens $W$, then automatically map relevant words to protected groups $G$. For example, if a model's explanation mentions women, this implies the protected group of gender. We construct a set of search words $S$ and map each one to at least one protected group $g \in G$. First, we gather a set of all protected groups from our definitions and manually map them to larger categories, as shown

in Table 9. Next, we gather all non-stopwords from the LLM responses. We search the words with ConceptNet (spe, 2017), and if there is a one-hop relationship between this word and one of those in Table 9, we add that to the searchable mapping. In this way, we obtain searchable keywords like young/old, men/women, christianity/judaism/nun, etc.

$$Jaccard(U, V) = \frac{|U \cap V|}{|U \cup V|} \qquad (1)$$

**Single-target vs. Multi-target.** A long-standing challenge in legal tasks is processing long, complex documents such as court opinions that often contain irrelevant information (Luo et al., 2023). To alleviate this issue, we refactor our classification to present one definition of hate speech at a time with a single-target configuration. For an input-output pair $(x, y) \in X$ with fine-grained labels $y = \{y_1, y_2, ..., y_n\}$ annotated to a set of definitions $D, |D| = n$, we refactor the output to create pairs $\{(x, y_1), (x, y_2), ..., (x, y_n)\}$. We prompt the model with each pair individually, then gather them back into one vector. For coarse-grained labels, we combine the decisions over 11 definitions by majority voting. Additionally, recent language models have been proposed with longer context windows (OpenAI, 2023), and we wish to evaluate their ability to reason over multiple targets. Where available, we compile all definitions into one prompt and ask the model to provide reasoning simultaneously.

## 4.2 Fine-tuning

We investigate fine-tuned language models to help establish baselines and provide insights for our task. In this work, we tune the encoder-only RoBERTa-Large (Liu et al., 2019) with parameter-efficient tuning over three settings:

- **LM-BFF** (Gao et al., 2021). We adopt the LM-BFF method as it allows us to prompt encoder-only language models in a similar format as autoregressive LLMs. This method takes the prompt text as input but appends a <mask> token to the end, converting the prompt response to a masked language modelling (MLM) objective. This method allows BERT to achieve similar performance to the original GPT-3.
- **Tuning** (T). We also tune RoBERTa-Large on the silver data with noisy labels generated by GPT-4. We freeze the RoBERTa-Large weights and only tune the encoder. This setting is chosen

to gain a sense of the dataset's difficulty without mentioning the definitions.

- **Self-training** (ST). With the silver data, we employ a self-training scheme to further improve performance. We use a simple self training routine, inspired by (Wang et al., 2021; Zou et al., 2019). With the best performing baseline checkpoint as the teacher, we generate inference pseudo-labels for our unlabelled data. Then, we use easy example mining (Kumar et al., 2010) to find 1,000 samples with the highest model prediction confidence. We train the checkpoint for 2 epochs as the student, then use the new model as the teacher and regenerate pseudo-labels. This process is repeated for $n$ rounds, until loss convergence. To compensate for noisy labels, we use label smoothing in the cross entropy loss.

## 5 Results and Discussion

### 5.1 Classification Results

**Coarse-grained Performance** Our experiment results are summarized in Table 3. The best performing model is GPT-4. Due to cost considerations, we were only able to run one round of experiments with GPT-4, so these results are considering the multi-target performance rather than single target. The two variants of GPT-3.5 perform similarly, but the 16k context variant performs slightly worse in fine-grained classification. Also, the 16k model exhibited 0.5% more hallucinations. Of the smaller models, Falcon-7b performs the best, while the macro-f1 scores for Alpaca-LoRA and WizardLM are comparable. Notice that Alpaca-LoRA has very minimal hallucinations, which suggests larger models are not strictly necessary to achieve low hallucination rates. However, lower hallucination rates are not indicative of higher performance, and there is a considerable performance gap between larger models, with over 100 billion parameters, and the smaller variants with less than 15 billion.

After self-training with silver data, coarse-grained performance of RoBERTa-Large improves significantly. This demonstrates the coarse-grained labels generated by GPT-4 are generally high quality. We observe RoBERTa-Large does not predict the Unclear class after self-training. The Unclear class is rare, especially after our positive class augmentation, so RoBERTa-Large can outperform GPT-3.5 by not predicting this class.

**Fine-grained Performance.** For fine-grained performance, there is a similar trend between larger

| Model | Coarse-grained | | | | | Fine-grained | | | | Fine-grained + reasoning | | | | |
|---|---|---|---|---|---|---|---|---|---|---|---|---|---|---|
| | Ma-f1↑ | Ma-P↑ | Ma-R↑ | Mi-f1↑ | HR↓ | Ma-f1↑ | Ma-P↑ | Ma-R↑ | Mi-f1↑ | Ma-f1↑ | Ma-P↑ | Ma-R↑ | Mi-f1↑ | HR↓ |
| GPT-3.5 | 30.7 | 46.5 | 52.5 | 27.1 | 0.05 | 45.4 | 30.0 | 96.5 | 45.3 | 31.8 | 19.1 | 99.7 | 32.3 | 0.6 |
| GPT-3.5-16k | 30.9 | 31.1 | 43.3 | 73.9 | 0.6 | 43.4 | 28.2 | 97.8 | 44.2 | 33.7 | 20.7 | 97.1 | 32.6 | 0.6 |
| GPT-4 | **41.4** | **38.1** | **48.0** | **92.5** | **0.04** | - | - | - | - | **52.2** | **43.9** | **67.7** | **55.1** | **0.04** |
| Falcon-7b | 17.4 | 25.1 | 24.6 | 39.7 | 14.0 | 12.1 | 7.0 | 46.1 | 12.1 | 10.0 | 6.0 | 31.6 | 10.2 | 63.4 |
| Alpaca-LoRA-7b | 21.1 | 28.6 | 34.7 | 46.4 | 0.06 | 15.1 | 8.4 | 87.1 | 14.8 | 10.5 | 5.6 | 92.9 | 10.9 | 21.7 |
| WizardLM-13b | 9.2 | 28.5 | 25.9 | 11.3 | 28.5 | 20.2 | 11.8 | 78.1 | 15.2 | 15.2 | 8.4 | 87.1 | 14.8 | 75.6 |
| Llama2-13b | 8.5 | 26.8 | 24.7 | 16.3 | 24.2 | 13.5 | 7.5 | 74.0 | 13.8 | 20.8 | 12.8 | 59.5 | 21.1 | 54.2 |
| RoBERTa-L (LM-BFF) | 5.4 | 35.8 | 33.7 | 8.5 | - | 8.8 | 4.8 | 62.7 | 9.1 | - | - | - | - | - |
| RoBERTa-L (T) | 1.1 | 33.3 | 0.6 | 1.7 | - | 0 | 0 | 0 | 0 | - | - | - | - | - |
| RoBERTa-L (ST) | 32.0 | 32.2 | 33.3 | 90.1 | - | 0 | 0 | 0 | 0 | - | - | - | - | - |
| Random | 22.6 | 30.8 | 33.3 | 46.3 | - | 10.6 | 6.0 | 51.0 | 10.7 | - | - | - | - | - |

Table 3: Summary of performance over our classification tasks. Results are single-target, except GPT-4 which reports one multi-target run. ↑ indicates a higher score is better, and ↓ indicates a lower score is better. Ma-f1, Ma-P, and Ma-R are Macro-f1, Macro-Precision and Macro-Recall respectively, HR is hallucination rate, in %. RoBERTa-L refers to RoBERTa-Large.

| Model | Coarse-grained | | Reasoning | |
|---|---|---|---|---|
| | Ma-f1↑ | Mi-f1↑ | HR↓ | J↑ |
| GPT-3.5 | 31.5 | 77.0 | 0.10 | 37.2 |
| GPT-3.5-16k | 31.3 | 76.6 | 0.04 | 36.4 |
| GPT-4 | 41.4 | 92.5 | 0.05 | 38.4 |

Table 4: Summary of our multi-target experiments, where we provide the entirety of our external legal reference to the model via prompt. Ma-f1 refers to Macro-f1, Mi-f1 is Micro-f1 score, HR is hallucination rate, and J is the Jaccard score for recognizing protected groups.

and smaller models, and GPT-4 still has the highest scores. It is interesting to note that GPT-3.5 and GPT-3.5-16k perform similarly, which indicates a longer context length does not necessarily lead to stronger understanding of the input. This agrees with previous work in legal applications that showed specialized encoder-only models like Longformer do not outperform BERT (Bhambhoria et al., 2021; Luo et al., 2023). Also, WizardLM outperforms the other open-source models in Macro-f1 score, and achieves double the performance of random guessing. While the GPT-series models seem to out-perform other works by a wide margin, the smaller models perform well relative to their size (7-13B parameters compared to 175B).

After fine-tuning RoBERTa-Large, the fine-grained performance reaches 0 due to model collapse. This is likely because our positive samples become too rare per definition for effective fine-tuning. We attempted various training strategies like upsampling the positive class, but the performance does not improve. In general, despite reaching similar coarse-grained performance, it seems like our task is difficult for smaller models to understand with naive training methods. We urge future works to further investigate this behaviour.

## 5.2 Reasoning Capabilities

**Explanations.** Next, we report the same experiments but prompt the model to explain its decision. When we ask the model to demonstrate its reasoning, the performance actually decreases on average. With additional investigation, we find that the models tend to hallucinate more when asked to provide an explanation — WizardLM, especially, begins to repeat the prompt back before generating the expected response. This might be because the WizardLM model is trained on highly complex tasks, while our instruction ("If yes, explain why.") was vague compared to the three-way Yes/No/Unclear decision. We sample outputs from different language models, presented in Table 5, and find that the quality of reasoning is not the same as our legal experts. Even GPT-4 makes frequent mistakes — the model identifies a violation of CC_318 (advocating genocide), but the statement and its own explanation do not reflect the severity of that charge. Legal texts often have complex and rare words, so a zero-shot LLM might benefit from adding definitions of words or other detailed instructions in-context.

**Multi-target Results.** Comparisons on GPT series models for long, multi-target prompts are summarized in Table 4. Due to cost considerations, we only collected multi-target results on GPT-4, so the results are identical compared to Table 3. The results of the two GPT-3.5 models improve, with GPT-3.5 dramatically improving in Micro-f1. The two GPT-3.5 models produce significantly fewer Unclear predictions in the multi-target setting. Per-

| Sample | Did he kill your sex life? Women are objects so you can just buy another one anyways. |
|---|---|
| Source | Response |
| GPT-3.5 | Yes, this post is a violation of TOS_RED. It promotes hate based on gender and objectifies women, which goes against the rule on hate and promoting attacks of hate. |
| GPT-4 | Yes, this post violates several of the above policies. 1. CC_318 and CC_319: This post can be seen as promoting hatred and disrespect towards women, which is a protected group under these policies. It devalues women by referring to them as objects that can be bought, which can be seen as advocating for their mistreatment... |
| WizardLM | Yes, this post is a violation of CC_318 as it promotes the idea that women are objects to be used and discarded, which is an act that falls under the definition of genocide. |
| Annotator | 'Wilful promotion of hatred', 'degrades women', 'The view of women as objects rises to the level of ardent and extreme feelings constituting hatred required to uphold the constitutionality of a prohibition of expression in human rights legislation', 'Hate speech and discriminatory against identifiable group' |

Table 5: A comparison of chosen language model outputs and our legal experts' notes for a gold data sample.

haps with a single definition, they are more likely to produce "neutral" responses to err on the side of caution. However, when at the limits of their reasoning capability, the models tend to be more decisive. Previously observed issues with hallucinations are also shown with these results. The Jaccard similarity is less than 0.5, indicating more than half the model's predictions and the gold annotations do not align on average. Here, the performance of GPT-4 is similar to both versions of GPT-3.5.

## 6 Conclusion

In this work, we introduce a new task formulation for hate speech detection grounded in enforceable legal definitions and release a gold label dataset annotated by legal experts. The strictness of these definitions results in a lower positive rate compared to previous annotated datasets. We augment the data with expert edits and automatic data augmentation. Then, we establish baseline performance on the state-of-the-art LLMs and tune encoder-based models. Finally, we tune with our silver data to further improve performance. We observe that LLMs perform impressively, but closer inspection of their explanations shows they lack reasoning. This work addresses the importance of legally enforceable hate speech, and we urge future work in this area.

## Limitations

Though the model is trained on legal definitions of hate speech, the application of these definitions to the facts remains subject to interpretation, and thus our outcome may not necessarily reflect the similar work trained on a different dataset. As observed when training on out of distribution data, our definition of hate speech is still incompatible with other works such as (Sachdeva et al., 2022). Our definition of hate speech could still be considered incomplete, as we do not consider definitions from the EU. As noted in section 3, nine of eleven definitions (including the five from social media companies) define very similar behaviours, to the point where we homogenize them in our second round of annotations. This might indicate our sampled definitions are culturally monotonous. In fact, hate speech is an inherently subjective concept, and we are aware of the biases our annotators hold in their decisions. For example, human rights codes and criminal law focus on defining protected groups, but violations are decided by a jury - laypeople - when a case is brought to court. We follow dataset annotation conventions, i.e. majority voting with opinions from at least three legal experts, to simulate a court decision.

Additionally, the baselines evaluated in this work have many limitations to be addressed before real-world implementation. Since we have limited training data, and mainly report zero-shot results, the performance and perspectives of the model are subject to the pre-training data. There are also issues inherent to the model, including hallucinations, lack of interpretability, among others. The data is not very suitable for tuning LLMs as well — the prompts are very monotonous, so the model can very easily start producing hallucinations.

## Ethics Statement

**Intended Use.** We see at least two applications for legal practice. First, this system could be used by social media companies to address or at least alleviate the issue of hate speech their platforms for hate speech. Since the annotations are aligned with legal definitions, such legal AI - provided it is aligned with the accepted legal interpretation of hate speech - might help the most egregious misconduct as well as offer some legal legal justification. Perhaps, its output might even be used as court evidence in court. Another application is an open access tool, such as add-on available on

website browser. Such tool might be used by public forum users who might have been victims of hate speech violations. They would input the text and the system would identify the relevant law(s), determining whether the statement constitutes hate speech.

**Failure Mode.** Due to the high stakes nature of the task, a false negative could lead to disastrous real-world violence against groups targeted by hate speech. Likewise, a false positive could lead to more severe consequences than someone deserves, which weakens user confidence and morale. However, the system is designed to be used to augment and assist human investigations, and these scenarios are unlikely to occur with sufficient human intervention.

**Misuse Potential.** As mentioned above, there is a chance for others to use these methods to completely automate regulation of their platforms and enact immediate consequences. Overconfidence in model predictions and insufficient understanding of the limitations of our work could lead to severe consequences in cases of failure. With the rising popularity of LLMs, it is important to further ground the model and prevent misuse.

## Acknowledgements

The research is in part supported by the NSERC Discovery Grants and the Research Opportunity Seed Fund (ROSF) of Ingenuity Labs Research Institute at Queen's University.

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

# A  Additional Data Details

## A.1  Legal definitions of Hate Speech

**Criminal Code.**  We use two sections of Canadian Criminal Code, Advocating Genocide and Public Incitement of Hatred. [4]

**Human Rights Codes.**  In Canada, human rights codes are maintained by provincial governments, and we select four that specifically mention hate speech: British Columbia, Alberta, Saskatchewan, and Northwest Territories. Other provinces' codes that mention discrimination without discussing hatred or contempt are not considered.

**Hateful Conduct Policies.**  The European Commission established a Code of Conduct in partnership with four social media companies to facilitate compliance with EU code, including adherence to the definition of hate speech defined by the EU (Commission et al., 2016). In addition, there is the Framework Decision on combating racism and xenophobia. The Framework Decision sets out minimum standards for the criminalization of racist and xenophobic speech and behavior, and requires EU Member States to adopt legislation criminalizing certain types of hate speech and hate crimes (of the European Union, 2008). Finally, the President von der Leyen announced in september 2020 the Commission's intention to propose to extend the list of EU crimes or Eurocrimes to all forms of hate crime and hate speech. We note that most social media companies also have more comprehensive standards for their content as drafted by their internal legal teams. We collect definitions from Twitter, Reddit, Youtube, Meta, and TikTok.

## A.2  Annotations

The annotators are ten law students (early-mid 20's) who volunteered for the project knowing their work might be used in academic publications. The students were either rewarded with credits as part of a practicum course, or paid the minimum wage in our region depending on the batch of annotations they participated in. For transparency, we did have one annotator that volunteered out of interest for the project, and they contributed <2% (<50/ 2100) of the annotations,  5 hours of work total (including meetings, training, and actual annotation work).

During annotation, they were provided a word document containing definitions and relevant ex-

---

[4] *Criminal Code*, R.S.C. 1985, c C-46, s. 318-319

| Forum | Count | Pos-o | Pos-l | Uncl. |
|---|---|---|---|---|
| CivilComments | 418 | - | 10 (2.4) | 17 (4.1) |
| Twitter | 39 | 14 (35.9) | 3 (7.7) | 5 (12.8) |
| Reddit | 16 | 11 (68.8) | 0 (0.0) | 2 (12.5) |
| Gab | 13 | 8 (61.5) | 9 (69.2) | 3 (23.1) |
| Stormfront | 2 | 0 | 0 | 1 (50.0) |
| Total | 488 | 39 (7.9) | 22 (4.5) | 28 (5.7) |
| with Augment. | 704 | - | 165 (23.4) | 38 (5.4) |

Table 6: Dataset distribution by source *forum*, as well as comparison of positive rates by previous hate speech definitions (Pos-o) to enforceable legal definitions (Pos-l). Uncl. refers to `Unclear` label rate, and all numbers are reported as Value (Rate %).

amples or caselaw, and the project was supervised by a law professor. First, annotators were asked to identify if the statement violates a legal definition of hate speech. If there is a violation, they were asked to indicate which definitions have been violated. Then, they were asked to explain their choice in free text form, and/or highlight free span text segments of the post they deemed most important to their decision. For posts where a majority voted positive, we pool the fine-grained definition labels into a set per sample. If at least one expert deems a definition has been violated, then it is assigned that fine-grained definition label.

We provide an `Unclear` label since we believe it is important for an automatic system to be able to recognize when a piece of text is ambiguous. While there are many clear statements of hateful speech, there are also times when the speaker's original meaning cannot be discerned, but the phrase contains hateful language or violent implications that causes concern, and a human expert believes it would require further context. The inter-annotator agreement, taken as Krippendorff's alpha, is 0.623 from the initial round of annotations, which shows a relatively high agreement. In the second round of annotations, we ask the annotators to identify the protected groups being targeted, infringements of the most severe cases (Criminal Code s. 318, 319), and generate fine-grained labels by processing the target.

### A.3 Additional Data Details

Positive class statistics by source forum are summarized in Table 6. Most of the data sources have fewer examples of legal hate speech compared to their original labels except Gab, where our positive rate is greater.

## B Additional Experimental Details

### B.1 Hyperparameters

We use the Adam optimizer with a learning rate of 5e-6, weight decay of 0.01, and training batch size of 8. We train for 10 epochs, which can take 10-15 hours for 4000 samples. When there are multiple target tasks available, we interleave datasets such that there is an even distribution during training. The open-source LLM experiments were performed on Nvidia 3090 GPUs with 24GB of RAM.

We use inference APIs from OpenAI for zero-shot LLM experiments, All OpenAI models are the versions released on June 13th, 2023, i.e. GPT-3.5 refers to model ID `gpt-3.5-turbo-0613`, GPT-16K-3.5 is `gpt-3.5-turbo-16K-0613`, and GPT-4 is `gpt-4-0613`. We use a decoding temperature of 0.1 and a top-p value of 0.75. For text preprocessing, we remove Unicode (except for " and ", which we replace with their plain text counterparts), anonymize any username references, and decode HTML strings (eg. & becomes &). We use the tiktoken[5] library to check the token length of all inputs, and we truncate text with length greater than 400 tokens.

---

[5] https://github.com/openai/tiktoken

Figure 4: The annotation interface shown to legal experts. This shows an example that was annotated as hate speech, but the definition options only appear after selecting "Violates."

| Legal source | *Criminal Code*, R.S.C. 1985, c C-46, s. 318 |
|---|---|
| **Prompt** | CC_318 is defined as Every person who advocates or promotes genocide is guilty of an indictable offence and liable to imprisonment for a term of not more than five years. In this section, genocide means any of the following acts committed with intent to destroy in whole or in part any identifiable group, namely, killing members of the group; or deliberately inflicting on the group conditions of life calculated to bring about its physical destruction. An intent to directly prompt or provoke another person to commit genocide is enough to establish the mens rea of the offence.

Protected/identifiable groups include colour, race, religion, national or ethnic origin, age, sex, sexual orientation, gender identity or expression, mental or physical disability

Post: The Indians need to purchase their own land and building. Canadians do not allow 6 year olds to have the temper tantrums that Indians inflict on Canadians.

Q: Is this post a violation of CC_318? Please answer yes or no. A: <mask> |

Table 7: An example of a full prompt in the template. Each model uses this template.

| | |
|---|---|
| Criminal Code s.318: Advocating Genocide | Age, **Religion/Religious beliefs**, **Sex/Gender identity or expression**, **Race (colour, creed)**, **Sexual orientation**, **Physical or mental disability**, **National or ethnic origin (nationality, ethnicity, ancestry)** |
| Criminal Code s.319: Public Incitement of Hatred | Age, **Religion/Religious beliefs**, **Sex/Gender identity or expression**, **Race (colour, creed)**, **Sexual orientation**, **Physical or mental disability**, **National or ethnic origin (nationality, ethnicity, ancestry)** |
| Human Rights Code, Alberta | Age, **Religion/Religious beliefs**, Source of income, Family status, **Sex/Gender identity or expression**, Marital status, **Race (colour, creed)**, **Sexual orientation**, **Physical or mental disability**, **National or ethnic origin (nationality, ethnicity, ancestry)** |
| Human Rights Code, British Columbia | Age, **Religion/Religious beliefs**, Family status, **Physical or mental disability**, **Sex/Gender identity or expression**, **Race (colour, creed)**, **Sexual orientation**, Marital status, **National or ethnic origin (nationality, ethnicity, ancestry)** |
| Human Rights Code, Northwest Territories | Age, **Religion/Religious beliefs**, Family status, Family affiliation, **Sex/Gender identity or expression**, Conviction that is subject to a pardon or record suspension, Marital status, Social condition, **Race (colour, creed)**, **Sexual orientation**, **Political belief/association**, **Physical or mental disability**, **National or ethnic origin (nationality, ethnicity, ancestry)** |
| Human Rights Code, Saskatchewan | Age, **Religion/Religious beliefs**, Family status, **Physical or mental disability**, **Sex/Gender identity or expression**, **Race (colour, creed)**, **Sexual orientation**, Receipt of public assistance, Marital status, **National or ethnic origin (nationality, ethnicity, ancestry)** |
| Terms of Service, Meta | **Religion/Religious beliefs**, **National or ethnic origin (nationality, ethnicity, ancestry)**, **Sex/Gender identity or expression**, Serious disease, **Race (colour, creed)**, **Sexual orientation**, **Physical or mental disability**, Family affiliation |
| Terms of Service, Reddit | **Religion/Religious beliefs**, **Sex/Gender identity or expression**, Victims of a major violent event and their families/kin, **Race (colour, creed)**, **Sexual orientation**, Immigration status, **Physical or mental disability**, **National or ethnic origin (nationality, ethnicity, ancestry)** |
| Terms of Service, TikTok | **Religion/Religious beliefs**, **National or ethnic origin (nationality, ethnicity, ancestry)**, **Physical or mental disability**, **Sex/Gender identity or expression**, **Race (colour, creed)**, **Sexual orientation**, Immigration status, Serious disease, Family affiliation |
| Terms of Service, Twitter | Age, **National or ethnic origin (nationality, ethnicity, ancestry)**, **Religion/Religious beliefs**, **Sex/Gender identity or expression**, Serious disease, **Sexual orientation**, **Race (colour, creed)**, **Physical or mental disability**, Family affiliation |
| Terms of Service, Youtube | Age, **Religion/Religious beliefs**, Family affiliation, **Sex/Gender identity or expression**, Victims of a major violent event and their families/kin, **Race (colour, creed)**, **Sexual orientation**, Veteran status, Immigration status, **Physical or mental disability**, **National or ethnic origin (nationality, ethnicity, ancestry)** |

Table 8: All definitions and their corresponding protected groups. Groups that are protected under all definitions are shown in bold.

| Race (colour, creed) | actual and perceived race, colour, coloured, black, creed, racially, races, race or perceived race |
|---|---|
| National or ethnic origin (nationality, ethnicity, ancestry) | ancestry, national, ethnic origin, ethnicity, place of origin, national origin, national or ethnic origin, nationality, indigenous identity |
| Political belief/association | political association, political belief |
| Sex/Gender identity or expression | gender, woman, men, transgender, boy, man, gender identity, gender identity and expression, gender identity or expression, sex, children, whore, sexist, baby, sex/gender |
| Religion/Religious beliefs | religion, cross, belief, christianity, judaism, faith, crosses, religions, nuns, religious affiliation, religion/religious, religious beliefs |
| Sexual orientation | sexual orientation |
| Social condition | social condition |
| Immigration status | immigration status, immigration |
| Source of income | source of income |
| Age | age, young, ages, old |
| Physical or mental disability | physical or mental disability, mental or physical disability, physical disability, mental disability, disability, impairment, disabilities, pregnancy or disability, adhd |
| Family affiliation | family affiliation, caste |
| Conviction that is subject to a pardon or record suspension | conviction that is subject to a pardon or record suspension |
| Receipt of public assistance | receipt of public assistance |
| Serious disease | serious disease, aids |
| Family status | family status |
| Pregnancy | pregnancy |
| Victims of a major violent event and their families/kin | victims of a major violent event and their kin, victims of a major event and their families |
| Veteran status | veteran status |
| Marital status | marital status |

Table 9: All search keywords used to identify target groups.