# OpenReview forum: "Legally Enforceable Hate Speech Detection for Public Forums"
_EMNLP/2023/Conference — EMNLP 2023 Findings_

### Official Review · Reviewer_G4KL · 2023-07-26

**Soundness:** 4

**Excitement:**

4: Strong: This paper deepens the understanding of some phenomenon or lowers the barriers to an existing research direction.

**Paper Topic And Main Contributions:**

The paper introduces a dataset for hate speech detection, labeled based on legal
definitions, and tests LLMs to assess their classification quality on them. A
frequent problem with hate speech datasets is the hate speech definition
difference across various corpora leading to various issues, such as
difficulties in enforcing the outputs of classifiers legally which the paper
aims to improve.  The dataset is composed of previously introduced datasets,
however, the input samples are relabeled by legal experts based on various legally
enforceable hate speech definitions, such as on federal laws. The authors
evaluated various LLMs on the database, including coarse- and fine-grained
classification setups, as well as the evaluation of model explanations,
serving as a benchmark for future research on the dataset.


**Questions For The Authors:**

* A: Hate speech definitions, including legal ones, evolve over time. I'm
wondering how well will this dataset stand the test of time? Is there a simple
way of keeping the dataset up to date instead of re-annotation by legal experts?
* B: Although Table 1 contains a comparison of the number of positive samples in
the original and the relabeled corpora, it would be interesting to discuss the
differences of the original definitions and the ones used in this work in more
details. What are the main differences of definitions other than the use of more
subtle forms of hate speech as mentioned in line 314?
* C: What is the degree of the annotators' agreement?


**Reasons To Accept:**

* The built dataset will be useful for researchers to evaluate classifiers
following hate speech definitions based on legally enforceable definitions.
* The paper also presents a useful benchmark for comparison of future systems,
which includes various models, model setups (zero-shot, tuning) and evaluation
setups. Additionally, the presented experiments provide a useful analysis on the
performance and behaviour of LLMs on legally enforceable hate speech detection.
* The writing of the paper is high quality and detailed.


**Reasons To Reject:**

* I did not find any significant reasons for rejecting the paper. Please see
below for some of my concerns/issues.


**Reproducibility:**

4: Could mostly reproduce the results, but there may be some variation because of sample variance or minor variations in their interpretation of the protocol or method.

**Reviewer Confidence:**

3: Pretty sure, but there's a chance I missed something. Although I have a good feel for this area in general, I did not carefully check the paper's details, e.g., the math, experimental design, or novelty.

**Typos Grammar Style And Presentation Improvements:**

* The description of the definitions in Table 8 is not presented. However, I
would suggest including them as well for more clarity.
* The description of WizardLM seems to be missing.
* There is a mismatch in the title between the pdf and Openreview.

---

> ### Author Rebuttal · Authors · 2023-08-25
>
> Thank you for taking the time to review our work! We will update your presentation improvements in the camera ready version of the paper. Please find our response to your questions below.
>
> ## Questions For The Authors
> **A: Hate speech definitions, including legal ones, evolve over time. I'm wondering how well will this dataset stand the test of time? Is there a simple way of keeping the dataset up to date instead of re-annotation by legal experts?**
>
> This is a valid concern for any hate speech dataset, and a primary motivation of this work. One advantage of using legal definitions is that they are normally passed after a rigorous evaluation process. They can be overturned or amended one day, but they have already been validated by many experts, and are relatively slow to change compared to previous works. We also analyze all 11 definitions and find that nine of them are relatively similar, as discussed in lines 223-231, so this dataset is in line with current western ideas of hate speech.
> We argue that our zero-shot experiments also show performance on unseen definitions of hate speech. We use unseen tokens (CC_318, CC_319, etc.) to encourage models to rely on the prompted definitions.
>
> **B: Although Table 1 contains a comparison of the number of positive samples in the original and the relabeled corpora, it would be interesting to discuss the differences of the original definitions and the ones used in this work in more details. What are the main differences of definitions other than the use of more subtle forms of hate speech as mentioned in line 314?**
>
> By “subtle” we refer to micro-aggressions and implicit statements, which are the main focus of Implicit Hate Speech and CounterContext. Actually, we argue these works are not aligned to the legal concept of hate speech, instead capturing the broader concept of _discrimination_ (prejudice towards certain groups of people). We internally discussed the enforceability of discrimination versus hate speech at great lengths before removing discrimination policies from our definition. Discrimination is a broad concept and its impact on society is varied, making the legal implications not well defined.
>
> (!!!!! WARNING: OFFENSIVE LANGUAGE)
> For example, here is one (cherry-picked) label-flipped sample from Implicit Hate Speech :
> > “this is crazy. how do you give a loan to someone who can be deported at will ?”
>
> The speaker believes “someone who can be deported at will” (immigrants) should not be able to get loans, which is discrimination but not hate speech. Hate speech in the law is defined as directly insulting/dehumanizing/humiliating someone based on a protected attribute. Multiple definitions reference, for example, comparing a group of people to insects.
>
> This might be closer to hate speech:
> > "they shouldn't be giving out loans to someone who can be deported at will. those cockroaches can’t be trusted"
>
> This is also a main motivation of our work — because previous works do not reference the law, they can conflate different legal concepts, and our understanding of hate speech in the NLP community becomes misaligned with experts in other disciplines.
>
> Thank you for the question. We will add this discussion to the paragraph at line 314 in the camera ready version of the paper.
>
> **C: What is the degree of the annotators' agreement?**
>
> The Krippendorff’s Alpha is 0.623, which indicates high agreement. This is mentioned on line 997 (Appendix A.2).

---

### Official Review · Reviewer_Ukeu · 2023-07-31

**Typos Grammar Style And Presentation Improvements:** N/A
**Soundness:** 3

**Excitement:**

3: Ambivalent: It has merits (e.g., it reports state-of-the-art results, the idea is nice), but there are key weaknesses (e.g., it describes incremental work), and it can significantly benefit from another round of revision. However, I won't object to accepting it if my co-reviewers champion it.

**Missing References:**

N/A

**Paper Topic And Main Contributions:**

The authors introduce a novel approach to hate speech detection by grounding the task in enforceable legal definitions. The authors present a dataset annotated by legal experts, containing violations of eleven possible definitions of hate speech. To address the challenge of identifying clear, legally enforceable instances of hate speech, the dataset is augmented with expert-generated samples. They report results on several large language models (LLMs) and demonstrate that with this task definition, hate speech detection can be more closely aligned with enforceable laws, thus enabling more rigorous enforcement of legal protections against harmful speech in public forums.


**Questions For The Authors:**

N/A

**Reasons To Accept:**

1. The paper introduces a unique perspective on hate speech detection, which focuses on grounding the task in legally enforceable definitions.
2. The release of a gold label dataset annotated by legal experts is a substantial contribution to the research community. Such a dataset can serve as a reliable benchmark for evaluating and comparing hate speech detection systems.


**Reasons To Reject:**

The paper primarily focuses on aligning hate speech detection with legal definitions. It may lack a broader analysis of other aspects related to hate speech and its impact.

**Reproducibility:**

3: Could reproduce the results with some difficulty. The settings of parameters are underspecified or subjectively determined; the training/evaluation data are not widely available.

**Reviewer Confidence:**

3: Pretty sure, but there's a chance I missed something. Although I have a good feel for this area in general, I did not carefully check the paper's details, e.g., the math, experimental design, or novelty.

---

> ### Author Rebuttal · Authors · 2023-08-25
>
> Thank you for your feedback on our work! Please find our response below.
>
> ## Reasons To Reject
>
> **The paper primarily focuses on aligning hate speech detection with legal definitions. It may lack a broader analysis of other aspects related to hate speech and its impact.**
>
> We argue that legal definitions have been created by policy makers and legal professionals who have analyzed and seek to mitigate the societal impacts of hate speech, so our work is already a direct assessment of hate speech and its impact. Previous works might capture various aspects of hate speech, but as discussed in lines 51-74, their definitions are often incomplete or inconsistent with one another.
> In general, we observed that existing works can often conflate concepts with hate speech that are distinct in the law, such as discrimination. As seen in Table 1 and discussed in lines 310-317, works that target more “subtle” definitions such as Implicit Hate Speech had all of their positive samples flipped to negative because of this conflation. For more details on differences between existing works and legal definitions, please refer to our rebuttal to reviewer G4KL.
>
> We believe that the community should shift their approach to align with existing, accepted policies of high impact tasks. This guides future research towards the interests of society and creates opportunities for applications and inter-disciplinary collaboration.

---

### Official Review · Reviewer_9nW2 · 2023-08-03

**Soundness:** 3

**Excitement:**

3: Ambivalent: It has merits (e.g., it reports state-of-the-art results, the idea is nice), but there are key weaknesses (e.g., it describes incremental work), and it can significantly benefit from another round of revision. However, I won't object to accepting it if my co-reviewers champion it.

**Justification For Ethical Concerns:**

No clear indication about annotator rewards

**Paper Topic And Main Contributions:**

This paper experiments with employing LLMs for legally enforceable hate speech. They have used existing datasets and incorporated definitions from current laws, in Canada. They have experimented with prompt engineering and fine-tuning LLM with existing data. They have also produced gold and silver datasets.

One big issue with this work is that the annotation/annotator strategy is not clear. If annotators are not rewarded for their task, this might raise an ethical issue.

**Questions For The Authors:**

.061
This subjectivity is not specific to computer sci-062
ence; hate speech is a highly debated topic, with ==> what do you mean by Computer science here?


Our dataset contains samples from existing datasets 240
with the aim ...==> Which existing dataset?

We employ editing annotation tasks and data fil-246
tering to increase the size of our gold data, while247
utilizing experts to ==> This is not clear? what do you employ?

It is not clear how the SBIC dataset is labeled (positive??)

HateXplain is not only from Twitter, but from Gab as well. Be precise in you related work

**Reasons To Accept:**

- Extend hate speech with prompt templating
- zero-shot classification for legally defined hate speech with many LLMs
- Fine0tuned models for legally enforceable hate speech
- Model result explanation

**Reasons To Reject:**

- How do we extend the approaches to other (countries' legal documents)?
- Data collection and annotation are not clear
- The Enforceable Annotation might have ethical issues. What will be the reward for the 10 law experts? why did they volunteer? Does it count toward their study (credit), or will they co-author the paper? This is a serious issue, which might lead to the low quality of the data.
- The concept of "editing samples" is not clear
- Majority voting from 11 definitions is not clear?
- You could compare your result with SoTA approaches, for example with HateXplain models.

**Reproducibility:**

4: Could mostly reproduce the results, but there may be some variation because of sample variance or minor variations in their interpretation of the protocol or method.

**Reviewer Confidence:**

5: Positive that my evaluation is correct. I read the paper very carefully and I am very familiar with related work.

**Typos Grammar Style And Presentation Improvements:**

search introduced in (Khurana 133
et al., 2022; Zufall et al., 2022)  ==> search introduced in Khurana 133et al., (2022); Zufall et al., (2022)

shot methods reduce requirements for training 142
data AlKhamissi et al. (2022). ==> ...Please correct the citation styles, it is very random. This should be as .... shot methods reduce requirements for training data (AlKhamissi et al. 2022).

---

> ### Author Rebuttal · Authors · 2023-08-25
>
> Thank you for taking the time to provide your feedback on our work! We will update the citation styles for the final version of the paper. Please find our response to your concerns below.
>
> ## Reasons To Reject
> 1. **How do we extend the approaches to other (countries' legal documents)?**
>
> Five of our definitions (Hateful Conduct Policies) are taken from social media companies, which can be applied internationally wherever these platforms are available.
>
> This concern is valid for unseen criminal law and human rights codes in other countries, as cross-cultural notions of hate speech may vary. As noted in lines 223-231, nine of eleven definitions (including the five from social media companies) define very similar behaviours, to the point where we homogenize them in our second round of annotations. This might indicate our sampled definitions are culturally monotonous. We will add this discussion to the limitations section of our work, but we do not believe cultural bias detracts from the value of our study.
>
> While there is likely cultural bias in both the language models and our dataset, we argue that our zero-shot experiment results can also generalize to unseen definitions of hate speech. As seen in Table 7, we use unseen tokens for the target (CC_318, CC_319, etc.) to encourage models to rely on the prompted definitions. We will add this detail to our description of Table 2 in the main paper content, and we will release the full annotation document and the per-definition prompts with the dataset upon acceptance.
>
> 2. **Data collection and annotation are not clear**
>
> We made best effort to describe the data collection and annotation process in Section 3.1 (lines 240-317), and provide additional details in Appendix A. We also make note to refer to the appendix from the main text (lines 197, 276, 305).
>
> 3. **The Enforceable Annotation might have ethical issues. What will be the reward for the 10 law experts? why did they volunteer? Does it count toward their study (credit), or will they co-author the paper? This is a serious issue, which might lead to the low quality of the data.**
>
> Thank you for your careful feedback. We take ethical concerns seriously and we thank you for noticing we failed to specify annotator compensation.
>
> The students were either rewarded with credits as part of a practicum course, or paid $15/hour, the minimum wage in our area, depending on the batch of annotations they participated in. For transparency, we did have one annotator that volunteered out of interest for the project, and they contributed <2% (<50/~2100) of the annotations, ~5 hours of work total (including meetings, training, and actual annotation work). We also ensured the data is high quality — as mentioned on line 997, the inter-annotator agreement is 0.623, indicating high agreement. We will add these details to Appendix A.2, line 970.
>
> 4. **The concept of "editing samples" is not clear**
>
> Please kindly refer to lines 334-340 for our description. “Editing” refers to adding or replacing words. To paraphrase the example we provide in line 339, “you are a jerk” is not hateful, but replacing “jerk” with a slur might be considered hateful.
>
> 5. **Majority voting from 11 definitions is not clear?**
>
> Please kindly refer to lines 301-305, as well as Appendix A.2, lines 983-986. We first take the coarse-grained label of Positive, Negative, or Unclear. If the sample is voted positive by a majority of 3 annotators, we take the or-wise combination of definitions/protected groups chosen by all annotators.
>
> 6. **You could compare your result with SoTA approaches, for example with HateXplain models.**
>
> Thank you for your comment. The main focus of our work is defining the task and establishing a baseline, so we believe it is not necessary to report state of the art approaches. In the HateXplain paper, they also report three general architectures: LSTMs, RNNs, and fine-tuned BERT.
>
> That being said, we do report results on the current state of the art in instruction-tuned LLMs (GPT-4), and we use the LM-BFF method to enable prompting with an encoder-only model, RoBERTa-Large. While these approaches are not specific to hate speech, they achieve high performance on instruction-based tasks similar to our experimental setting.
>
> ## Questions For The Authors
> 1. **Clarifying line 61-62 reference to computer science**
>
> We are referring to works in NLP and computational linguistics that seek to create automatic hate speech detection systems, in contrast with works in social sciences or law that debate the concept of hate speech; we will clarify this sentence in the final version of the paper.
>
> 2. **Clarifying datasets, line 239-240**
>
> We sample from all non-enforceable datasets listed in Table 1 and described in lines 251-283.
>
> 3. **Clarifying lines 245-247, methods to increase gold data**
>
> We employ the editing objective, lines 318-355, and automatic data selection, lines 356-387, to increase the size of the gold dataset.
> It is not clear how the SBIC dataset is labeled (positive??)
>
> 4. **Clarifying labels of SBIC**
>
> “Positive” refers to a positive sample of hate speech, following our naming convention of positive, negative, and unclear. As described in lines 262-265, we use a heuristic of offensive speech targeted at a group, both of which are existing annotations in the SBIC. Formally, we consider annotations where (offensiveYN > 0.5 && whoTarget == group) as the positive class, following previous work.
>
> 5. **HateXplain description**
>
> Thank you for the feedback. We acknowledge this mistake in our description, and we will update the text to say “Twitter and Gab” in the camera-ready version of our paper. Our dataset does include 13 HateXplain samples from Gab, as seen in Table 6 of Appendix A.3.

---

### Meta-Review · Area_Chair_9L4w · 2023-09-12

**Recommendation:** 4

**Metareview:**

The paper focusses on legally enforceable hate speech detection.
The most important contributions is the release of a dataset annotated by legal experts.
The paper also presents a useful benchmark for comparison of future systems, which includes various models, model setups (zero-shot, tuning) and evaluation setups.

The criticism that the paper "lacks a broader analysis of other aspects related to hate speech and its impact." is somewhat irrelevant to the main contribution it is making.

---

### Decision · Program_Chairs · 2023-10-07

**Decision:**

Accept-Findings

**Comment:**

The paper focusses on legally enforceable hate speech detection.
The most important contributions is the release of a dataset annotated by legal experts.
The paper also presents a useful benchmark for comparison of future systems, which includes various models, model setups (zero-shot, tuning) and evaluation setups.

The criticism that the paper "lacks a broader analysis of other aspects related to hate speech and its impact." is somewhat irrelevant to the main contribution it is making.